# Enhancement of Antiviral CD8^+^ T-Cell Responses and Complete Remission of Metastatic Melanoma in an HIV-1-Infected Subject Treated with Pembrolizumab

**DOI:** 10.3390/jcm8122089

**Published:** 2019-12-01

**Authors:** Oscar Blanch-Lombarte, Cristina Gálvez, Boris Revollo, Esther Jiménez-Moyano, Josep M. Llibre, José Luís Manzano, Aram Boada, Judith Dalmau, Daniel E. Speiser, Bonaventura Clotet, Julia G. Prado, Javier Martinez-Picado

**Affiliations:** 1IrsiCaixa AIDS Research Institute, 08916 Badalona, Spain; oblanch@irsicaixa.es (O.B.-L.); cgalvez@irsicaixa.es (C.G.); ejimenez@irsicaixa.es (E.J.-M.); jdalmau@irsicaixa.es (J.D.); bclotet@irsicaixa.es (B.C.); 2Autonomous University of Barcelona, 08193 Cerdanyola del Vallès, Barcelona, Spain; aramboada@gmail.com; 3Infectious Diseases Department, University Hospital “Germans Trias i Pujol”, 08916 Badalona, Spain; brevollo@flsida.org (B.R.); jmllibre@flsida.org (J.M.L.); 4Medical Oncology Service—Badalona Applied Research Group in Oncology (B-ARGO Group), University Hospital “Germans Trias i Pujol”—Catalan Insitute of Oncology (ICO), 08916 Badalona, Spain; jmanzano@iconcologia.net; 5Department of Dermatology, University Hospital “Germans Trias i Pujol”, 08916 Badalona, Spain; 6Department of Oncology, University of Lausanne, CH-1015 Lausanne, Switzerland; 7Chair in Infectious Diseases and Immunity, Faculty of Medicine, University of Vic-Central University of Catalonia (UVic-UCC), 08500 Vic, Spain; 8Germans Trias i Pujol Research Institute (IGTP), 08916 Badalona, Spain; 9Catalan Institution for Research and Advanced Studies (ICREA), 08010 Barcelona, Spain

**Keywords:** Immune checkpoint inhibitors, pembrolizumab, HIV-1 reservoir, HIV-specific CD8^+^ T cells, HIV-1 curative strategies

## Abstract

Background: Pembrolizumab is an immune checkpoint inhibitor against programmed cell death protein-1 (PD-1) approved for therapy in metastatic melanoma. PD-1 expression is associated with a diminished functionality in HIV-1 specific-CD8^+^ T cells. It is thought that PD-1 blockade could contribute to reinvigorate antiviral immunity and reduce the HIV-1 reservoir. Methods: Upon metastatic melanoma diagnosis, an HIV-1-infected individual on stable suppressive antiretroviral regimen was treated with pembrolizumab. A PET-CT was performed before and one year after pembrolizumab initiation. We monitored changes in the immunophenotype and HIV-1 specific-CD8^+^ T-cell responses during 36 weeks of treatment. Furthermore, we assessed changes in the viral reservoir by total HIV-1 DNA, cell-associated HIV-1 RNA, and ultrasensitive plasma viral load. Results: Complete metabolic response was achieved after pembrolizumab treatment of metastatic melanoma. Activated CD8^+^ T-cells expressing HLA-DR^+^/CD38^+^ transiently increased over the first nine weeks of treatment. Concomitantly, there was an augmented response of HIV-1 specific-CD8^+^ T cells with TNF production and poly-functionality, transitioning from TNF to an IL-2 profile. Furthermore, a transient reduction of 24% and 32% in total HIV-1 DNA was observed at weeks 3 and 27, respectively, without changes in other markers of viral persistence. Conclusions: These data demonstrate that pembrolizumab may enhance the HIV-1 specific-CD8^+^ T-cell response, marginally affecting the HIV-1 reservoir. A transient increase of CD8^+^ T-cell activation, TNF production, and poly-functionality resulted from PD-1 blockade. However, the lack of sustained changes in the viral reservoir suggests that viral reactivation is needed concomitantly with HIV-1-specific immune enhancement.

## 1. Introduction

Inhibitory receptors (iRs) or immune checkpoint molecules play a key role in the regulation of persistent immune activation during cancer and chronic infections to avoid self-damage. Immune checkpoint inhibitors (ICIs) are monoclonal antibodies that target iRs to reverse T-cell exhaustion due to continuous antigen stimulation. Pembrolizumab (Keytruda^®^, MSD) is an ICI directed against programmed cell death protein-1 (PD-1), which blocks the interaction with its ligands PD-L1 and PD-L2. These ligands are overexpressed on activated antigen presenting cells as well as cancer cells, and their blockade promotes T-cell activation against tumor cells [1]. Currently, pembrolizumab, approved by FDA, has become a first-line treatment against metastatic or unresectable melanoma providing a five-year survival rate between 34% and 41% [2].

Expression of PD-1 has also been associated with dysfunctional HIV-1 specific-CD8^+^ T-cell responses and disease progression. In this context, in vitro PD-1 blockade has demonstrated a recovery of HIV-1 specific-CD8^+^ T-cell functionality [3,4]. On the other hand, iR-expressing CD4^+^ T cells have been associated with cell-based measures of viral persistence in HIV-1-infected patients on antiretroviral therapy (ART) [5,6], and ex vivo experiments have shown an enhancement of viral production by CD4^+^ T cells after PD-1 blockage [7,8]. Therefore, ICIs could impact on the HIV-1 reservoir through the so-called ‘shock and kill’ mechanism; i.e., reactivating latent HIV-1 provirus from infected CD4^+^ T cells and reversing exhausted HIV-1 specific-CD8^+^ T cells against HIV-1 producing cells, altogether resulting in a potential reduction of the reservoir that ultimately might help viral remission [9].

Different case reports, and a couple of case series, suggest that ICI therapy appears to be safe and efficacious in HIV-1-infected individuals with advanced stage cancer [10], although ongoing prospective trials of ICI need to confirm these findings [11,12,13]. However, the effectiveness of ICI therapy to help eliminate the viral reservoir in ART-treated individuals is still controversial and only based on limited case reports. These cases range from showing no major changes in either HIV-1 specific-CD8^+^ T-cell responses or HIV-1 persistence [14], transient enhancement of HIV-1 specific-CD8^+^ T cells with no variation in viral persistence [15,16], transient increase in viral transcription without changes in viral reservoirs [17], or one case of depletion of the HIV-1 reservoir [18].

Here, we report a case of an HIV-1-infected individual on ART that received pembrolizumab treatment for metastatic melanoma. We combine the description of clinical response to melanoma after pembrolizumab treatment with a detailed characterization of functional T-cell responses as well as the therapeutic impact on viral persistence.

### Case Report

A 46-year-old man was diagnosed with HIV-1 infection in 2008, immediately receiving suppressive ART (Figure 1A). In 2011, his ART was simplified to ritonavir-boosted darunavir. In June 2016 he was diagnosed with an amelanotic melanoma on the right hemithorax, with palpable right axillary lymphadenopathy. Staging positron emission tomography-computed tomography (PET-CT) and neurologic magnetic resonance image (MRI) showed no evidence of distant metastatic disease. In September 2016, an axillary lymphadenectomy was performed with involvement of 3/22 lymph nodes by melanoma. There were no mutations in the codon V600 of the BRAF gene, NRAS in exon 2 or 3, or in the KIT gene (PCR Sanger sequencing, Therascreen BRAF Pyro Kit, Qiagen; Hilden, Germany). Repeated PET-CT showed disease progression with hypermetabolic focal lesions in the right axillary, right pleura, and D4 vertebral body (Figure 1B1–B3). In October 2016, the subject initiated pembrolizumab (2 mg/kg every three weeks). At the onset of melanoma, the ART was switched to raltegravir 400 mg twice daily and tenofovir fumarate/emtricitabine to avoid drug interactions. His CD4^+^ T-cell count was 544 cells/µL and plasma HIV-1 RNA was suppressed (<40 copies/mL). The subject had never presented any AIDS-associated illness. Genotyping of HLA class I loci indicated homozygosity for HLA-A*02, HLA-B*07, and HLA-C*07. HLA class II loci showed homozygosity for HLA-DRB1*15, and HLA-DQB1*06.

A new PET-CT after 12 cycles of pembrolizumab treatment showed complete disappearance of the right axillary, right pleura, and D4 vertebral body lesions (Figure 1B4–B5). The infusions were well tolerated, without requiring systemic steroids for signs of local inflammation to pembrolizumab.

Peripheral blood mononuclear cells (PBMCs) and plasma samples were collected before each pembrolizumab administration at weeks (w) 0, 3, 9, 18, 27, and 36 (Figure 1A). Additionally, we evaluated the short-term effect of pembrolizumab in a sample one day after pembrolizumab administration at w18 (w18+1). The subject provided informed consent and the institutional review board approved the investigational protocol (ref PI-18-229).

## 2. Experimental Section

### 2.1. Immune Phenotype of Total CD8^+^ and HIV-1 Specific-CD8^+^ T Cells

To monitor changes in CD8^+^ T cells and HIV-1 specific-CD8^+^ T-cell responses after pembrolizumab initiation, PBMCs were stimulated with HIV-1 Gag peptide pool (2 μg/peptide/mL, EzBiolab, Carmel, IN, USA), Staphylococcal enterotoxin B (SEB) (1 μg/mL, Sigma-Aldrich, Madrid, Spain), or no stimuli, in the presence of CD28/49d co-stimulatory molecules (1 μg/mL, BD, Madrid, Spain), Monensin A (1 μg/mL, BD), and CD107a (PE-Cy5, clone eBioH4A3, BD) for 6 h at 37 °C in a 5% CO_2_ incubator and rested overnight at 4 °C. After incubation, cells were washed with PBS 1X and stained with a viability dye (APC-Cy7, Thermo Fisher Scientific) for 30 min at room temperature (RT). Cells were surface stained for 30 min at RT with anti-human antibodies for CD3 (A700, clone UCHT1, BD), CD4 (APC-Cy7, clone SK3, BD), CD8 (V500, clone RPA-T8, BD), CD45RA (BV786, clone HI100, BD), CCR7 (PE-CF594, clone 150503, BD), CD27 (BV605, clone L128, BD), PD-1 (BV421, clone EH12.1, BD), CD38 (PE, clone HB-7, BD), and HLA-DR (A647, clone L243, BioLegend, London, UK). Afterwards, cells were fixed with Fix/Perm Buffer A (Thermo Fisher Scientific) for 15 min at RT and intracellular stained with Fix/Perm Buffer B and antibodies for TNF (PE-Cy7, clone MAb11, BioLegend), IFNγ (BV711, clone B27, BD), and IL-2 (BV650, clone MQ1-17H12, BD) for 20 min at RT. Finally, cells were resuspended and fixed in formaldehyde 1% and acquired on LSR Fortessa cytometer using FACSDiVa software (BD). Data were analyzed with FlowJo software v10. We excluded dump and CD4^+^ T cells, and gates were defined using fluorescence minus one (FMO) controls. Surface markers were measured in total CD8^+^ T cells and CD8^+^ T-cell subsets including naïve (T_N_), central memory (T_CM_), transitional memory (T_TM_), effector memory (T_EM_), and effector (T_E_), as previously described [19]. The frequency of each surface marker was expressed as the mean of two independent replicates. For HIV-Gag specific-CD8^+^ T-cell cytokine production, only positive replicates were included. GraphPad Prism v4.0 software was used for graph plotting. Pestle and Spice software [20] was applied to visualize poly-functionality of HIV-1 specific-CD8^+^ T-cell responses for the evaluation of cytokine pattern distribution.

### 2.2. Total HIV-1 DNA, Cell-Associated HIV-1 RNA, and Ultrasensitive Viral Load

To evaluate the size of the proviral reservoir, total HIV-1 DNA was measured in purified peripheral CD4^+^ T-cell lysates using droplet digital PCR (ddPCR), as previously described [21]. Briefly, 5′ long terminal repeat (5′LTR) region was amplified in duplicate, and the RPP30 housekeeping gene was quantified in parallel to normalize sample input. Raw ddPCR data were analyzed using the QX100™ Droplet Reader and the QuantaSoft v.1.6 software (Bio-Rad, Hercules, CA, USA).

Also, viral transcription was evaluated by quantification of cell-associated HIV-1 RNA (ca-HIV-1 RNA) in purified CD4^+^ T cells, using one-step reverse-transcription ddPCR22. The 5′-LTR gene and the housekeeping gene TATA-binding protein (TBP) were measured in parallel. Residual low-level viremia was determined through ultrasensitive viral load (usVL) by ultracentrifugation of 5–6 mL of plasma and quantified using the Abbott Real-Time HIV-1 assay (Abbott Molecular Inc., Des Plaines, IL, USA), as previously described [22,23]. All determinations were done on equivalent number of CD4^+^ T cells.

## 3. Results

### 3.1. Pembrolizumab Administration Induces a Marked and Transient Increase of HLA-DR^+^/CD38^+^ Expression in T_EM_ and T_E_ CD8^+^ T Cells

Immunotherapies are supposed to affect the distribution of immune cell populations and their activation status. We measured changes in the frequency of total CD8^+^ T cells (Figure 1C) and stable counts were observed despite a transient decrease at w9. The frequency of total CD8^+^ T cells was 68.5% at baseline and was contracted to 62.5% at w9, returning to baseline levels beyond w18. Regarding CD8^+^ T-cell subpopulations (Figure 1D), no major changes were observed in the distribution over the analyzed period. Only the expression of T_N_ and T_TM_ showed a transient increase between w9 and w18. When we evaluated the short-term effect at w18+1, we identified a slight decrease in total CD8^+^ T cells (70.9% to 67.4%) that was concomitant in T_N_, T_CM_, and T_TM_ subpopulations. However, T_E_ and T_EM_ raised 7.3% and 7.2%, respectively, after 24 h of pembrolizumab infusion (Appendix A).

Concerning the activation status, total CD8^+^ T cells showed increased expression of the activation markers HLA-DR and CD38 after the first pembrolizumab infusion [24,25] (Figure 1E). Activated CD8^+^ T cells raised progressively from w0 to w9 (12.7% to 23.6%) and returned to baseline levels at w18 (12.7%). This transient increase of HLA-DR^+^/CD38^+^ expressing cells was also observed in all the CD8^+^ T-cell subpopulations. The percentage of activated T_N_, T_TM_, T_EM_, and T_E_ increased gradually after the first infusion and peaked at w9 (Figure 1F). Particularly, the expression of HLA-DR^+^/CD38^+^ in T_EM_ and T_E_ cells doubled from 14.2% to 27.4% and from 13.9% to 25.4%, respectively. Nevertheless, activated T_CM_ cells peaked at w3 and their levels decreased progressively overtime. Regarding the short-term effect at w18+1, we observed a small increase in the activation status of total CD8^+^ T cells (12.7% to 13.9%) but no changes in CD8^+^ T-cell subsets (Appendix A). Altogether, these data show that pembrolizumab transiently activates peripheral CD8^+^ T cells, more specifically effector cells, within the first few infusions in line with its immune therapeutic mechanism.

### 3.2. Enhancement of TNF Production in HIV-1 Specific-CD8^+^ T-Cell Responses after Pembrolizumab Initiation

In addition to the evaluation of changes in CD8^+^ T-cell populations and their activation status, we wanted to measure the magnitude and functional profile of HIV-1 specific responses. Thus, we performed a detailed characterization of HIV-1 specific-CD8^+^ T cells by a combination of surface lineage markers and intracellular cytokine detection in response to HIV-Gag. We found an increase in HIV-1 specific-CD8^+^ T-cells defined by an augment of TNF production between w0 and w9 (0.3% to 0.5%) after the first infusion of pembrolizumab (Figure 2A). The response decreased progressively to undetectable levels at w36. Interestingly, concomitant to TNF reduction, we observed a slight increase of IFNγ, CD107a, and IL-2 production by HIV-1 specific-CD8^+^ T cells at w18, which stayed that way until the end of the follow-up. Additionally, at w18+1 we observed an increase from 0.2% to 0.4% on TNF production by HIV-1 specific-CD8^+^ T cells. However, no changes were observed in CD107a, IFNγ, and IL-2 (Appendix A). Also, we observed a high frequency of TNF, CD107a, IFNγ, and IL-2 cytokine-producing cells in response to CD8^+^ T-cell polyclonal activation with the SEB stimuli that transiently peaked at w3 (Figure 2B).

By using lineage expression markers, we tried to decipher the phenotype that underlined the peak of TNF production by HIV-1 specific-CD8^+^ T cells. We observed a decrease in CCR7, CD27, and an increase in HLA-DR and PD-1 markers between w0 and w9 (Figure 2C). These data indicate an effector profile of HIV-1 specific-CD8^+^ T cells responsible of TNF production at the peak of response to pembrolizumab.

### 3.3. Functional Switch from TNF to IL-2 Profile in HIV-1 Specific-CD8^+^ T-Cell Responses after Pembrolizumab Follow Up

By using a detailed functional profile of cytokine production based on TNF, IFNγ, CD107a surface expression, and IL-2 expression, we analyzed the evolution of HIV-1 Gag-specific CD8^+^ T-cell responses overtime (Figure 2D,E). Pembrolizumab raised the cell’s polyfunctionality, expressed as any combination of two or more cytokines, from 28.7% at w0 to 78.2% at w36 (Figure 2D). Regarding combinations of three cytokines, although we observed peaks at w3 (24.9%) and w9 (22.3%), these combinations remained high until the end of the study. Indeed, detailed analyses showed a transient increase driven by HIV-1 specific-CD8^+^ T-cells expressing TNF (TNF^+^) during the first nine weeks of pembrolizumab treatment. Nonetheless, the TNF dominated profile switched to an IL-2 dominant profile (IL-2^+^ and IL-2^+^IFNγ^+^) since w9 (Figure 2E). In the short-term effect, we found an augment of TNF expressing cells (TNF^+^) at w18+1, concomitant with a reduction in IL-2^+^ production (Appendix A). However, the combination IL-2^+^IFNγ^+^ seems to start to raise at this timepoint. These data contrast with those from polyclonal stimuli where no changes were observed. Thus, the initial release of TNF by HIV-1 specific-CD8^+^ T cells is switched to an IL-2 release profile in the long-term antiviral response.

### 3.4. Transient Decays of Total HIV-1 DNA

To monitor the perturbation in the HIV-1 CD4^+^ T-cell reservoir induced by pembrolizumab, we measured viral persistence by HIV-1 DNA, ca-HIV-1 RNA, and usVL. For all the analyzed samples, we detected total HIV-1 DNA with a median IQR of 1008 (862–1141) copies/million CD4^+^ T cells. We detected transient decays in total HIV-1 DNA from 1080 to 820 copies per million CD4^+^ T cells (24%) at w3, and 727 copies per million CD4^+^ T cells (32%) at w27 (Figure 3A). However, after each reduction, the levels of total HIV-1 DNA recovered to values similar to those of the baseline. Moreover, the expression of cell-associated unspliced HIV-1 RNA in CD4^+^ T cells was stable over time in all samples with a median level of 40 (36–47) ratio (HIV/TBP) x 1000 (Figure 3A).

In order to determine the effect of pembrolizumab on HIV-1 reactivation, we measured HIV-1 RNA by ultrasensitive viral load. We detected HIV-1 RNA in 50% of analyzed plasma samples. Interestingly, rather than seeing the production of new virions, we observed a reduction of viremia below the limit of detection (0.8 copies/mL) at w3 and w27 (Figure 3B). This decrease in HIV-1 RNA was concomitant to the transient reduction of total HIV-1 DNA observed. When we evaluated the short-term effect of pembrolizumab at w18+1, we found levels of total HIV-1 DNA and ca-HIV-1 RNA within the same magnitude than at w18 (Appendix A). Thus, we found transient decreases of total proviral HIV-1 DNA and ultrasensitive plasma HIV-1 RNA at w3 and w27 after pembrolizumab treatment without affecting cell-associated unspliced HIV-1 RNA in CD4^+^ T cells.

## 4. Discussion

Previous studies have hypothesized that ICIs might be possible candidates to the “shock and kill” curative strategy by the reactivation of the HIV-1 reservoir [7,8,26] and the concomitant enhancement of antiviral immune responses [3,4,27], leading to the clearance of reactivated cells and consequently reducing the HIV-1 reservoir size. However, the effect of PD-1 blockade in HIV-1-infected individuals is incompletely understood, and the results are controversial, in part due to limited experience in this clinical setting. Here, we not only show a case of complete remission of metastatic melanoma after treatment with pembrolizumab in a HIV-1-infected individual, but also an enhancement of T-cell activation and the HIV-1 specific-CD8^+^ T-cell response. However, this T-cell invigoration was associated only with a transient perturbation on the HIV-1 reservoir size.

Pembrolizumab had a good safety profile and a sustained antitumoral response in this HIV-1-infected subject. Although the experience of pembrolizumab, as well as other ICIs, in people with HIV-1 is limited, these results are in line with other preliminary studies assessing the safety of ICI therapies in HIV-1-infected individuals with advanced-stage cancer [10,11]. However, only a fraction of these patients with advanced melanoma benefit from ICI blockade [10,11], which highlights this case of complete remission of this metastatic melanoma. Immunologically, over the first nine weeks, we found an increase of HLA-DR^+^/CD38^+^ expressing cells, especially in T_EM_ and T_E_ CD8^+^ T-cell subsets, which had also been reported previously [18]. While previous studies only found IFNγ production by HIV-1 specific-CD8^+^ T cells [15,16,18], we describe for the first time the production of a peak of TNF production by HIV-1 specific-CD8^+^ T cells at w9 as a result of the PD-1/PD-L1 pathway disruption by pembrolizumab, in agreement with the initiation of an inflammatory response. TNF is a pleiotropic cytokine, mainly produced by activated cells, regulating the immune response, promoting inflammation, and inhibiting tumorigenesis [28].

After the peak of TNF, we observed a transition from TNF to an IL-2 cytokine profile consistent with the reduction of T-cell activation through the antigen receptor [29]. IL-2 plays a crucial role in the expansion of effector T cells that get activated upon antigen encounter, and regulates proliferation and homeostasis of T, B, and NK cell responses [28,30]. IL-2 helps to resolve the inflammatory response caused by pembrolizumab in the first weeks of treatment, contributing to the maintenance of HIV-1 specific-CD8^+^ T cells and generating a long-term memory response [30]. However, we cannot exclude an impact of pembrolizumab on other cell types including NK cells. Indeed, NK cell responses can have crucial immunotherapeutic effects upon PD-1/PD-L1 blockade in tumors with low levels of MHC-I expression [31]. Additional studies profiling functional immune cell responses during treatment with ICIs could bring new insights on the mechanism of these new therapies in the remission of cancer and viral diseases.

Despite observing two punctual drops on proviral DNA during the study, we were unable to detect a consistent decrease in the viral reservoir. This observation has been previously reported by other studies using ICIs in combination with ART [14,15,16,17]. These results differ from a single case report that shows a decrease from 369 to 30 copies HIV-1 DNA per million cells at day 120 after nivolumab administration [18]. Also, it has recently been reported that pembrolizumab led to transiently short-term small increase in unspliced cell-associated HIV-1 RNA in CD4^+^ T cells in vivo in individuals on ART [32]; however, due to differences in sampling times, we cannot corroborate such observations.

The absence of sustained reduction on proviral DNA in our study case might be due to the lack of expression of cell-associated HIV-1 RNA and the subsequent production of viral antigen needed for the targeting of infected cells by effector HIV-1 specific-CD8^+^ T cells [26]. Although there was an enhancement on HIV-1 specific-CD8^+^ T-cell responses, we did not observe a sustained reduction of viral reservoir in peripheral CD4^+^ T cells, suggesting that pembrolizumab was unable to effectively reverse viral latency. Alternatively, any sustained reduction in the viral reservoir might have been unnoticed due to homeostatic proliferation of latently HIV-1-infected CD4^+^ T cells expressing PD-1, which might mask the detection of any effective “shock and kill” events under pembrolizumab treatment. Furthermore, the limitation of our analyses to peripheral blood samples underestimate the effects on lymphoid tissues where CD4^+^ T follicular helper cells expressing PD-1 play an important role as HIV-1 reservoirs [33]. Furthermore, data from ex vivo measurements on CD4^+^ T cells from ART-suppressed individuals suggest that pembrolizumab enhances the induction of HIV-1 production when combined with the latency reversing agent bryostatin, without providing data in reservoir clearance [8].

This study is restricted to peripheral blood samples from a single patient with complete control of viral replication under suppressive ART, which might have masked any beneficial effect of anti-PD1 treatment on plasma HIV viral load. These limitations should be carefully considered, in view of interindividual variability, especially in cancer patients. Ongoing clinical trials administrating ICI alone [32,34] or in combination with latency reversing agents to HIV-1-infected individuals on ART would help to clarify the potential contribution of ICIs in curative strategies.

Overall, our study shows that pembrolizumab treatment resulted in a successful remission of metastatic melanoma in a HIV-1-infected individual, and concomitantly enhanced the HIV-1 specific-CD8^+^ T-cell response in terms of activation and function. Nevertheless, this was not sufficient to consistently reduce the size of the viral reservoir, questioning the stand-alone effect of ICIs in terms of “shock and kill” HIV-1 curative strategies.

## Figures and Tables

**Figure 1 jcm-08-02089-f001:**
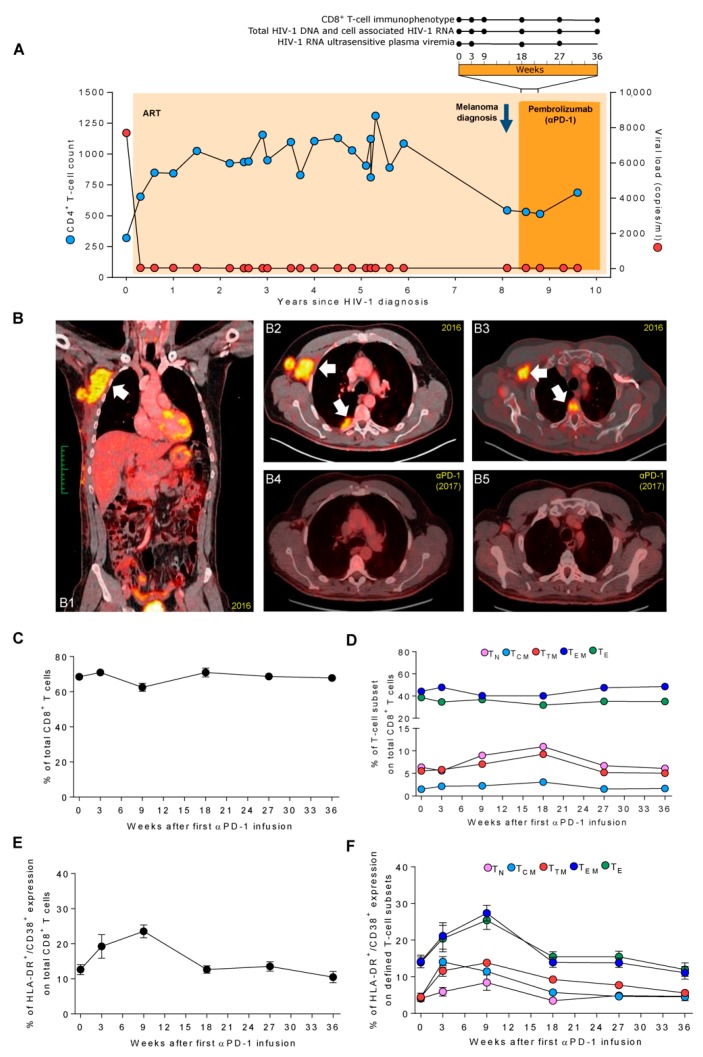
Clinical follow-up and impact of pembrolizumab administration in an HIV-1-infected individual on active antiretroviral treatment (ART) who developed metastatic melanoma; (**A**) Longitudinal analysis in years of CD4^+^ T-cell count (blue circles) and HIV-1 viral load (red circles) since HIV-1 diagnosis. Light orange area indicates the time on ART and dark orange area depicts the time of pembrolizumab administration under ART. Blue arrow indicates melanoma diagnosis. Top bar represents the cycles of pembrolizumab administrations and the biological samples that were analyzed at 0, 3, 9, 18, 27, and 36 weeks. Black lines indicate measures in CD8^+^ T-cell immunophenotype, total HIV-1 DNA, cell-associated (CA) HIV-1 RNA and, ultrasensitive viral load in plasma (HIV-1 RNA (usVL); (**B**) coronal (left panel) and axial (right panel) images of positron emission tomography with 2-deoxy-2-(fluorine-18) fluoro-d-glucose integrated with computed tomography (18F-FDG PET/CT). **B1**–**B3** images depict the lesions before pembrolizumab administration (27 September 2016) by the increased uptake of 18F-FDG in the right axillary lymphoid node and local soft tissue invasion in pleura and osteoblastic spine metastasis. **B4** and **B5** panels show serial imaging after one year of pembrolizumab administration (27 September 2017) of axial baseline and indicating sustained complete metabolic response and the resolution of the lesions. For **B1** to **B5**, white arrows indicate the localization of the metastatic lesions of melanoma; (**C**) changes in total CD8^+^ T cells after the first pembrolizumab administration in weeks; (**D**) changes in CD8^+^ T-cell subsets; (**E**) analysis of total CD8^+^ T cells expressing HLA-DR^+^/CD38^+^; and (**F**) CD8^+^ T-cell subsets expressing HLA-DR^+^/CD38^+^. T_N_, naïve; T_CM_, central memory; T_TM_, transitional memory; T_EM_, effector memory; T_E_, effector. Results are expressed as the mean of two measurements. RAL, Raltegravir; TDF/FTC, truvada.

**Figure 2 jcm-08-02089-f002:**
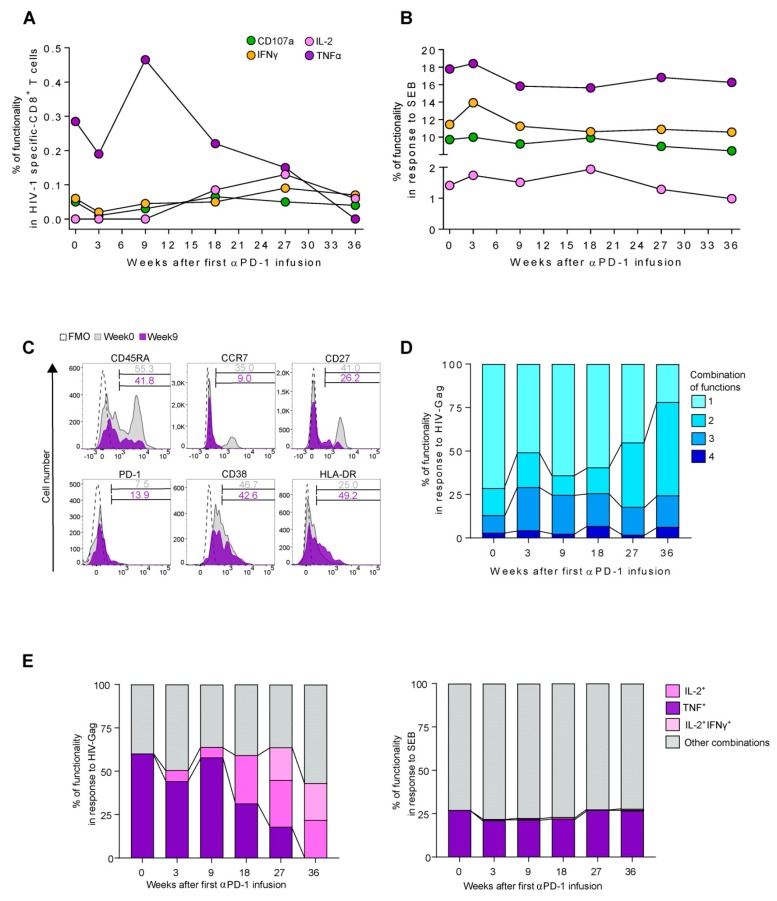
Immunophenotype of the HIV-1 specific-CD8^+^ T-cell response during pembrolizumab administration; (**A**) The graph represents the changes of TNF, IFNγ, CD107a, and IL-2 expression in Gag HIV-1 specific-CD8^+^ T-cells responses; (**B**) cytokine changes in response to Staphylococcal enterotoxin B (SEB); (**C**) histograms depicts the expression of CD45RA, CCR7, CD27, PD-1, CD38, and HLA-DR on TNF HIV-1 specific-CD8^+^ T cells at week 0 (grey) and week 9 (purple). Fluorescence minus one controls (FMOs) for each marker are included; (**D**) poly-functionality expressed as the combination of one, two, three, or four cytokines in HIV-1 specific-CD8^+^ T cells; (**E**) detailed poly-functional profiles of HIV-1 specific-CD8^+^ T cells and under SEB stimulation. The bars in purple represent the evolution of TNF^+^ frequency overtime in single combination and pink represents the changes of IL-2^+^ and IL-2^+^IFNγ^+^ over the week 18.

**Figure 3 jcm-08-02089-f003:**
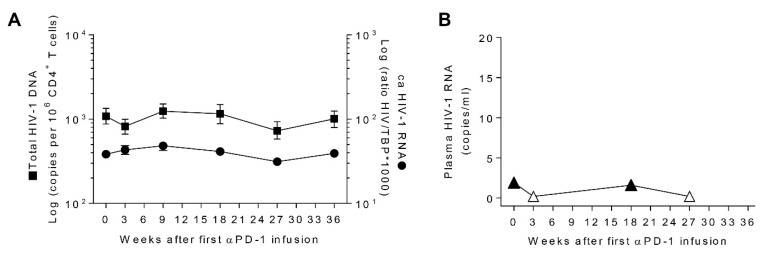
Longitudinal analysis of HIV-1 reservoir during pembrolizumab administration; (**A**) Total HIV-1 DNA (squares) and cell-associated HIV-1 RNA (circles) in CD4^+^ T cells, measured by ddPCR; (**B**) ultrasensitive viral load in plasma (triangles). Open symbols represent determinations below the limit of quantification.

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
