# Peer review of "Enhancement of Antiviral CD8+ T-Cell Responses and Complete Remission of Metastatic Melanoma in an HIV-1-Infected Subject Treated with Pembrolizumab"

_jcm, 2019, doi:10.3390/jcm8122089_

Round 1

Reviewer 1 Report

The authors describe a case of a patient with HIV who also had melanoma with ganglion dispersion. Anti-PD1 treatment prescribed for melanoma resulting in total remission.

The authors study the effect that melanoma treatment to inhibit the immune checkpoint has on the response of CD8 lymphocytes against the HIV virus and on the circulating viral load and reservoir in CD4 lymphocytes.

The authors describe a discrete effect of PD1 treatment on CD8 lymphocytes, with a small increase in their number and a transient production of TNF that does not affect circulating viral load or viral reservoir in CD4 lymphocytes.

Data collected in the literature are contradictory about whether anti-PD1 treatment has any effect on the course of HIV infection. For this reason the authors study in depth the modifications that occur in CD8 lymphocytes and their effect on viral load.

The results provided do not clarify the situation, possibly because the patient is controlled with his antiretroviral treatment and under these circumstances is very difficult to observe appreciable changes. The methods used are suitable for the study.

1.- The authors should compare the clinical situation of the HIV patient under study with other cases in the literature. Perfect clinical control of infection could mask beneficial effects of anti-PDF treatment on HIV viral load.

2.- The authors consider as active CD8 lymphocytes those expressing CD38/HLADR. This statement should be indicated in the literature.

Minor point: The study on HIV virus reservoir in CD4 lymphocytes should indicate whether all determinations are based on an equivalent number of CD4 cells.

Author Response

Point 1: The authors should compare the clinical situation of the HIV patient under study with other cases in the literature. Perfect clinical control of infection could mask beneficial effects of anti-PD1 treatment on HIV viral load.

Response 1: We had already compared the clinical situation of this case with previously published case reports in terms of safety profile and cancer benefit (lines 269-274). All cases described in the scientific literature, including the one that showed a reduction in cell-associated HIV-DNA (Guihot et al. 2018) reported clinical control of HIV infection as a result of effective antiretroviral therapy. We have now included a sentence indicating that effective control of HIV replication under antiretroviral therapy might mask the beneficial effects of anti-PD1 treatment on HIV viral load (lines 314-316.

Point 2: The authors consider as active CD8 lymphocytes those expressing CD38/HLADR. This statement should be indicated in the literature.

Response 2: The expression of activation antigens, HLA-DR and CD38, on circulating HIV-specific CD8 lymphocytes during HIV-1 infection has been widely used since the early 90’s. We have included a couple of supporting references (ref 24 and 25, line 179). Nevertheless, we analyzed the data exclusively using the HLA-DR marker on CD8 T cells with identical dynamics.

Minor point 3: The study on HIV virus reservoir in CD4 lymphocytes should indicate whether all determinations are based on an equivalent number of CD4 cells.

Response 3:  Longitudinal determinations were done on equivalent numbers of CD4+ T cells. Also, the quantification was normalized by sample input by using housekeeping genes (RPP30 or TBP depending on the measurement) (line 164).

Reviewer 2 Report

It is an interesting study by Blanch-Lombarte et al., who show enhancement of polyfunctional anti HIV-1 CD8+ T cell responses in a patient treated with anti PD1 immunotherapy. Most experiments are well controlled to the extent and limits of human patient.

It is surprising that even under these rejuvenated conditions the viral reservoirs are hardly perturbed.

Specific points

TNFalpha should be changed to TNF They can consider including HLA of the patent so that it would be easier to compare with past as well as future studies of patients treated with anti PD-1 in the context of HIV infection The polyfunctionality of CD8+ T cells shown using pie charts (Fig 2E) is extremely complicated and hard to grasp.

Authors should consider including a simplified version of these data on the line of Fig 2D, using bar graphs, which are easier to grasp and compare. Showing progressive loss of TNF and emergence of IL2 producing profile.

Author Response

Point 1: TNFalpha should be changed to TNF.

Response 1: As suggested we have changed TNFα for TNF through all the manuscript.

Point 2: They can consider including HLA of the patent so that it would be easier to compare with past as well as future studies of patients treated with anti PD-1 in the context of HIV infection

Response 2: We have included the HLA of the subject in the clinical case report section (line 97-99).

Point 3: The polyfunctionality of CD8+ T cells shown using pie charts (Fig 2E) is extremely complicated and hard to grasp. Authors should consider including a simplified version of these data on the line of Fig 2D, using bar graphs, which are easier to grasp and compare. Showing progressive loss of TNF and emergence of IL2 producing profile.

Response 3: As suggested we have changed Fig. 2E and supplementary Fig. 1E using bars instead of pie charts to facilitate data interpretation.